# Transitioning to a Low-Carbon Lifestyle? An Exploration of Millennials' Low-Carbon Behavior—A Case Study in China

Yan Wu [1,*], Pim Martens [2] and Thomas Krafft [1]

1 Department of Health, Ethics & Society, CAPHRI Care and Public Health Research Institute, Faculty of Health, Medicine and Life Sciences, Maastricht University, P.O. Box 616, 6200 MD Maastricht, The Netherlands; thomas.krafft@maastrichtuniversity.nl

2 System Earth Science, University College Venlo, Faculty of Science and Engineering, Maastricht University, P.O. Box 616, 6200 MD Maastricht, The Netherlands; p.martens@maastrichtuniversity.nl

* Correspondence: yan.wu@maastrichtuniversity.nl

**Abstract:** The Sustainable Development Goals (SDGs) have set the agenda for 2030, calling for collective global efforts to deal with climate change while seeking a balance between economic development and environmental protection. Although many countries are exploring emission reduction paths, mainly from government and corporate perspectives, addressing climate change is also an individual responsibility and requires public participation in collective action. The millennial generation constitutes the current workforce and will be the leaders in climate action for the next 30 years. Therefore, our study focuses on the Chinese millennial generation, conducting in-depth semi-structured interviews with 50 participants in qualitative research to explore their low-carbon lifestyles, the barriers, and enablers in switching to a wider range of low-carbon lifestyles. There are three main results: (1) Based on our study samples, there is an indication that Chinese millennials have a positive attitude towards transitioning to a low-carbon lifestyle. Women demonstrate a stronger willingness to adopt low-carbon behaviors in their daily household activities compared to men. However, their involvement in governance in the context of transitioning to a low-carbon society is limited, with most women assuming execution roles in climate action rather than decision-making positions. (2) Millennial's low-carbon life transition is accompanied by technological innovation and progress. However, this progress brings some new forms of resource waste, and reasonable policy-making is essential. (3) Personal economic interests and the satisfaction of their consumption needs will drive millennials to reduce carbon emissions in their daily lives, but it requires the guidance of reasonable policy-making and synergies among various stakeholders. This research will help policymakers better understand the current status and potential issues related to people's low-carbon actions, enabling the formulation of more rational guiding policies. It can also help other stakeholders learn about millennials' demands and take more effective collective action toward carbon reduction.

**Keywords:** climate change; low-carbon behavior; millennials' lifestyle; city transformation; China; qualitative research

## 1. Introduction

July 2023 is considered the hottest month on record for the Earth's temperature [1]. Some locations in the USA, China, Europe, and Africa were hit by a record-breaking heatwave. For example, Phoenix, Arizona in America recorded 31 days, as of 30 July, of daytime temperatures above 43.3 °C [2]; on 16 July, Turpan City in China's Xinjiang province, had a temperature of 52.2 °C [3,4], which would have been about a 1 in 250-year event [2]. Extreme heat poses serious threats to people's health and the environment [3]. This alarming trend underscores the urgent need for concerted action toward mitigating climate change.

Low-carbon transition goals were integrated into the global development agenda [5]. In the past decades, governments have announced their reducing carbon emission goals,

which are in line with the Paris Agreement [6]. For example, the European Union aimed to realize climate neutrality by 2050 and achieve an economy with net-zero greenhouse gases [7]; Germany set the goal of becoming net zero emissions by 2045 [8]; China aimed to reach peak $CO_2$ emissions by 2030 and achieve carbon neutrality before 2060. However, the current climate change positions of various countries are often based on their economic interests. We are halfway to 2030, and despite the passage of seven years, we have made little progress toward achieving the Sustainable Development Goals (SDGs); thus, the next seven years will be the key [9]. Therefore, this article believes that coping with climate change is not only the responsibility of governments but also of individuals. It requires civil society, businesses, and others to work collectively [10]. This paper focuses on exploring the attitudes and behaviors of millennial stakeholders towards low-carbon initiatives and the potential for collaborative efforts in reducing carbon emissions.

Urban areas contribute to over 70% of $CO_2$ emissions and are major contributors to climate change [11]. China's urbanization rate has exceeded 65%. Reducing urban greenhouse gas emissions is crucial for China to achieve its peak carbon and carbon neutrality goals [12]. Achieving carbon neutrality requires companies, organizations, or individuals to reduce their total greenhouse gas (GHG) emissions over a certain period. Energy conservation and emission reduction will be helpful in ultimately reaching "zero emissions" [13]. Previous studies on low-carbon cities in China, from the research content perspective, primarily focused on the development or impact of low-carbon policy from conceptual theory [14–19], the discussion of pilot low-carbon city development patterns through case studies [20–24], and carbon emission calculation and energy consumption in the industry [25–28]. Citizens are the most important part of cities; they are energy users and consumers in cities and have the responsibility to take climate action and join low-carbon activities. However, there is limited research focusing on the public's low-carbon behavior and attitude toward low-carbon lifestyles in China.

Previous studies on low-carbon cities in China, from the research method perspective, most utilized quantitative research based on statistical analysis. For example, Zhang and Zheng [29] used a penal data analysis to show that pilot low-carbon city policy can promote urban residents to transfer to a green lifestyle. Some researchers calculated residents' household carbon emissions through questionnaire surveys [30,31], while others employed the difference-in-differences model based on experimental evidence to explore the impact of pilot low-carbon city policies on urban carbon intensity [32–34]. While quantitative analysis excels in measuring and analyzing large datasets to provide clear numerical evidence, it may overlook crucial qualitative aspects such as personal experiences, emotions, and psychology that influence human behavior [35,36]. Some complex social phenomena are not easy to quantify. Low-carbon city transformations encompass various aspects, including policy-making, environmental education, public participation, and social equity. However, there is inadequate literature utilizing qualitative research to explore the potential synergistic and incentive policies in low-carbon city transformation in China. This research will address this gap, using a semi-structured interview method within a qualitative framework to examine individuals' attitudes toward a low-carbon society transformation as well as their willingness to reduce carbon emissions and participate in climate action.

Climate change is a challenge of collective action, not only within generations but also across generations [37]. Previous research indicates that age and education could affect people's environmental perception [38]. Millennials, the generation born between 1981 and 1996 (they were between 27–42 years old in 2023) [39], have already entered the job market, and they are poised to become the future workforce and shake things up across various fields [40]. Additionally, they bear the responsibility of raising and nurturing the next generation. Therefore, they play an important role in dealing with climate change and realizing carbon neutrality in the next 30 years. Petrescu-Mag et al. [41] used qualitative research to interview 41 Romanians, comparing millennials and Gen Z's attitudes toward health concerns regarding climate change in Romania. Some researchers used the questionnaire to explore millennials' attitudes towards climate change in America [42] or social media's

impact on individual behavior in South Korea [43]. There is limited research focusing on millennials' attitudes towards climate change or low-carbon transition in China.

Chinese millennials grew up with the policy of reform and opening up in 1978, and most of them are the only child in their families due to the one-child policy from the 1980s, which was loosened in 2015. Unlike previous generations in China, millennials have more opportunities to receive higher education in universities and colleges. The gross tertiary education rate in China increased from 2.7% in 1978, when the national college entrance examination (known as Gaokao) was resumed, to 59.6% in 2022 [44]. In the past 40 years, the growth of this generation was accompanied by China's rapid economic development. China's GDP growth has averaged over 9 percent per year since 1978, and over 700 million Chinese have been lifted out of poverty [45,46]. Due to disparities in education, economic conditions, and social culture, the younger generation's lifestyle diverges from that of their parents or grandparents [47]. They exhibit higher levels of indulgence, sophistication, and a propensity for consumption [48]. Although they are the beneficiaries of economic development, they are the bearers of environmental risks, and they will also be responsible for the mitigation of climate change in the future. There is significant potential for public participation in climate action in China. Existing research on public environmental awareness and green behaviors has primarily focused on the behaviors of teenagers in China [49–51], with a lack of studies targeting millennials. Lifestyle is regarded as individual behavioral patterns within the sociomedical discourse [52]. Embracing lifestyle change forms an essential element of transformative paths towards achieving net-zero emissions [53]. Given the unique context of millennials, this article seeks to explore low-carbon behaviors and lifestyles from their perspective in China. By understanding their motivations and challenges, we can foster meaningful engagement and accelerate progress toward a sustainable future. Three questions will be addressed in this research:

(1) What are the low-carbon behaviors of millennials in their daily lives?
(2) What are the barriers to a wider low-carbon lifestyle for millennials?
(3) What factors enable millennials to adopt a low-carbon lifestyle?

Using in-depth interviews within qualitative research, this research investigates millennials' low-carbon awareness and behaviors in low-carbon city transformation in China. It explores barriers to adopting low-carbon lifestyles and identifies motivating factors for engaging in climate action. This study contributes to the literature on low-carbon lifestyles and urban transitions in the following three innovative ways.

Firstly, we focus on the Chinese millennial generation, a demographic with 2–20 years of work experience transitioning from formal education to broader social learning. They often contribute significantly to carbon emissions as the main body of the consumption market and are more likely poised to assume managerial positions in various industries. Their adoption of low-carbon practices could influence organizational strategies, thereby impacting a wider range of people. This focus is different from previous studies, which primarily explore the low-carbon awareness and behavior of teenagers or college students in schools [54].

Secondly, we employ in-depth interviews to investigate respondents' low-carbon awareness and behaviors, as well as their perceptions of peer behavior. The sample extends previous studies' analysis quality by engaging stakeholders from diverse industry sectors. The flexible interview format enables us to delve beyond surface-level insights, exploring respondents' initial reactions, perceptions, and emotions. More importantly, we discussed in depth the underlying factors shaping their low-carbon choices. Therefore, our research overcomes the limitations of traditional survey methods within quantitative research prevalent in the existing literature [55,56].

Thirdly, we summarize and analyze the respondents' low-carbon behaviors in their lives, work, and social interactions, offering a holistic view beyond the focus on household-level carbon emissions often found in current research [31,57]. Our study not only advances the understanding of millennial attitudes towards low-carbon lifestyles but also enriches the methodology for studying such complex societal shifts.

The findings of this study can help policymakers improve their strategies for low-carbon urban transitions. Furthermore, our research will benefit stakeholders across various fields by improving their low-carbon products and services, thus contributing to building a more sustainable future through cooperation.

## 2. Method

### 2.1. Semi-Structured Interviews

In this study, we used semi-structured interviews within qualitative research to explore (1) millennials' low-carbon attitudes and behavior, (2) the barriers to switching to a low-carbon lifestyle, and (3) the enablers in turning to a low-carbon lifestyle for millennials. The semi-structured interview is widely used in qualitative research for its ability to elicit comprehensive insights into a topic of the daily world or social phenomenon [58–60]. This kind of interview collects information through conversation under the guidance of open-ended questions, and the researchers can deepen the questions according to the interviewee's answers [61,62]. We chose a semi-structured interview because it provides the flexibility needed to deeply understand the interviewees' first impressions and reactions to a low-carbon lifestyle from their unique perspectives. This method facilitates one-to-one conversations, which is better suited to learning about participants' mental processes [63].

A semi-structured interview is suitable for this research because, unlike questionnaires, it is more flexible in content and is not constrained by written language. This allows us to conduct in-depth investigations and explore various aspects of our research questions. Chinese millennials, as products of the one-child policy, grew up alongside China's rapid economic growth during the reform and opening-up period, with increased opportunities to receive higher levels of education and have distinctive growth experiences. They are oriented towards experiences, seeking individuality, and pursuing lifestyle upgrades [64]. However, they are also the main body of the consumption market, which causes huge amounts of carbon emissions [65]. This approach enables us to capture the complexities of participants' attitudes and behaviors, providing deep insights into their willingness to adopt and switch to a low-carbon lifestyle.

Since qualitative semi-structured interviews typically involve small sample studies, the purpose is to gain an in-depth understanding of participants' individual experiences rather than calculating the number of occurrences of an event or assessing the extent of an effect [66–69]. The dialog between researchers and participants is flexible rather than following pre-set questionnaires or scales [70]. Therefore, statistical analysis such as standard deviation and confidence interval analyses are not applicable in this context [71–75], as they are more aligned with probability inferences and error estimation in large-sample quantitative research [76]. Following the guidance of qualitative research, providing detailed method descriptions could enhance the reliability and effectiveness of the research results [77]. Therefore, this study provides detailed method descriptions about Sampling and Participant Recruiting in Section 2.2, Data Collection Processes in Section 2.3, and Data Analysis in Section 2.4. This study also introduces the potential limitations of this method in Section 5 to maintain the rigor of research.

### 2.2. Sampling and Participant Recruiting

We combined purposive sampling and snowballing methods to recruit participants. Purposive sampling was used to select different cases according to our research questions and aim. In qualitative research, scholars widely use purposive sampling techniques to identify participants who can potentially provide the most effective information based on their related knowledge or experience about the discussion phenomenon [78,79]. This method requires researchers to select cases with variation in order to collect diverse and rich information [80] to achieve a depth of understanding. Therefore, we first created a potential participant list based on our personal networks. Our research targets millennials who were aged 27–42 in 2023 (born between 1981 and 1996) who work in various organizations, including government agencies, academic institutions, media institutions, NGOs, residents, and business companies,

with a focus on different fields. Practitioners in these fields are stakeholders in the low-carbon city transformation [20], and they are also knowledgeable about millennials' lifestyles. This age group is 20–35 years away from retirement, and most of them have some work experience. Many people have played or will play managerial roles in their fields, and they will be the key figures over the next 30 years in China's pursuit of carbon neutrality by 2060. Therefore, millennials' awareness and behavior regarding low-carbon lifestyles will not only affect the education of the next generation but will also play a significant role in the low-carbon transition and decision-making within their organizations or industries.

In addition, the snowball sampling approach was also used to recruit participants because this method is effective in accessing difficult-to-reach or 'hidden' populations by leveraging the existing social connections of individuals with relevant characteristics [81–83]. Based on the first stage of conducting the preliminary interviews, we sought recommendations for additional interviewees who could also be suitable participants for this study. Moreover, we utilized social platforms such as LinkedIn, WeChat, and Weibo to identify and contact relevant interviewees from various sectors and fields. There were three selection criteria: (1) they were millennials aged between 27 and 42 in 2023 (born between 1981 and 1996); (2) they were familiar with low-carbon-related terms such as climate change, reuse, recycling, garbage sorting, etc.; (3) they know about low-carbon development within their fields.

Based on previous studies on sample size in qualitative research, many researchers adhere to the notion that data saturation, as defined by Glaser and Strauss, depends on the presence of new relevant information in subsequent interviews [84–86]. Sample size guidelines suggest a range of 20 to 30 interviews as sufficient [87–92]. Interview data collection is deemed complete when no further discoveries surface [91,93]. This should be distinguished from the larger sample size of written surveys [94] because the advantage of qualitative interviews lies in their interactive nature, their ability to capture the complexity of experience, and their allowance for unexpected topics to emerge, enriching the data due to their depth and duration [95]. In line with the research advice of Loraine, B. et al. [96], this study was first carried out through online interviews (through video calling and voice calling) with 21 participants between 1 August 2022 and 16 September 2022 due to the COVID-19 travel restrictions. Following the initial data collection, the subsequent step involved data analysis. To meet our research needs, we then carried out face-to-face interviews with nine participants from 1 February 2023 to 29 February 2023. Through this further data collection, we were able to identify new information during this phase of analysis. This iterative process persisted until no new information emerged. Redundancy of information occurred after we conducted an additional 20 online in-depth interviews in January and February of 2024 to enrich our data, reaching a point termed saturation. Each interview duration ranged from 30 min to 2 h, depending on whether the interviewees fully answered our questions or if there were any additional in-depth explorations related to our research topics and aims.

In total, we reached out to 74 interviewees; 50 individuals met our research criteria and were willing to participate in interviews, and the other 24 potential participants declined due to time constraints or a lack of related knowledge.

## 2.3. Data Collection

Semi-structured interviews can be conducted in person or online, and researchers can flexibly arrange the interview questions [60]. Before the interviews, we sent interview invitation letters to all potential participants, introducing details, including research aims, interview questions, privacy protection, data confidentially, and the information collection and data usage process. When participants agreed to participate, we made an appointment for an interview time slot with them. Before each interview, we also introduced the background of our research and reconfirmed the process and anonymity. The interview questions began with general questions about the participants' personal information, like age, education, and role in their institution. Then, the full interview was guided by four sections with 10 questions (Sheet S5 and Appendix A). The order of interview questions was arranged flexibly based on respondents' knowledge and background to facilitate deeper communication. All participants

were free to decide whether to participate and were informed they could abort the interview at any time. Interviews were recorded using the cell phone's recording function, and the interviewer made notes with important information, with the participants' agreement. To ensure the protection of participants' privacy and anonymity, we implemented a coding structure that used numerical identifiers instead of names in the transcripts. Table 1 shows the basic demographic information of the interviewees.

**Table 1.** Basic demographic information of interviewees.

| Characteristic | Demographic | Frequency | Percentage |
|---|---|---|---|
| Age | 27–29 | 14 | 28% |
| | 30–34 | 25 | 50% |
| | 35–39 | 8 | 16% |
| | 40–42 | 3 | 6% |
| Gender | Male | 24 | 48% |
| | Female | 26 | 52% |
| Education | Junior college | 2 | 4% |
| | Bachelor's degrees | 18 | 36% |
| | Master's degree | 25 | 50% |
| | Doctor's degree | 5 | 10% |
| Organizations | Government | 4 | 8% |
| | Traditional medias | 16 | 32% |
| | Corporates | 16 | 32% |
| | Educational institutions | 3 | 6% |
| | Residents | 9 | 18% |
| | NGOs | 2 | 4% |
| Areas | First-tier cities in China | 24 | 48% |
| | New first-tier cities in China | 16 | 32% |
| | Second or third-tier cities | 10 | 20% |

While interviews provide in-depth insights and researchers can capture participants' emotions, perspectives, and experiences, it is also important for researchers to observe the environment and context [97]. To gain a better understanding of the current urban lifestyle in China and examine the interviewees' responses, we conducted fieldwork as an additional approach during the in-person interview phase in Beijing, Hohhot, Nanjing, and Suzhou from 1 February 2023 to 29 February 2023. Our fieldwork involved observing waste-sorting practices in 20 communities, the utilization of shared bicycles and new energy vehicles in these cities, and a low-carbon company.

*2.4. Data Analysis*

The interviews were digitally recorded and fully transcribed. Our study follows an inductive approach based on grounded theory in qualitative research to analyze the data, focusing on obtaining synergistic knowledge and showing the empirical evaluations [98]. Data collection and analysis were conducted simultaneously, in line with the iterative approach of qualitative research [99,100], until no new information appeared to reach the data saturation [101]. After each interview, we promptly organized and analyzed the data. This research involved several analysis steps. Figure 1 shows the data analysis process of this research. We began by transcribing the interviews in Atlas.ti 22.2.0 qualitative software, and then reviewed all transcriptions twice to ensure the accuracy of the original interview data.

Next, we proceeded with the coding process. According to grounded theory, the coding process includes three phases. First is open coding. We used an inductive approach based on the text of the interviews to analyze data [102,103]. The transcripts were reviewed line-by-line, word-by-word, or phrase-by-phrase. Preliminary codes were generated based on the 'in vivo' terms used by the participants themselves after comparing for similarities and differences [104]. This process involved selecting paragraphs, fragments, and significant quotes from the transcripts [105]. Subsequently, after the initial coding, we checked and added the coding, if necessary, and removed duplicates.

The second phase is axial coding. This involves drawing connections between the initial open codes to form categories [106]. The acquired codes were compared with previous ones, and conceptually comparable codes were grouped together to form categories. In addition, categories were compared and, if necessary, combined. In some cases, one category was divided into two or more [107]. We organized the initial codes into fourteen categories based on their commonality and the textual content of the interviews [108].

Thirdly, selective coding, the conclusive step in grounded theory analysis, facilitates forming a larger theoretical scheme by integrating categories [109]. It aims to unveil the interrelations among categories to establish the ultimate thematic frameworks [110]. We organized the previously established categories into three overarching themes aligned with our research questions: millennials' low-carbon behavior, the barriers to a low-carbon lifestyle, and the possible enablers for millennials to live a low-carbon life.

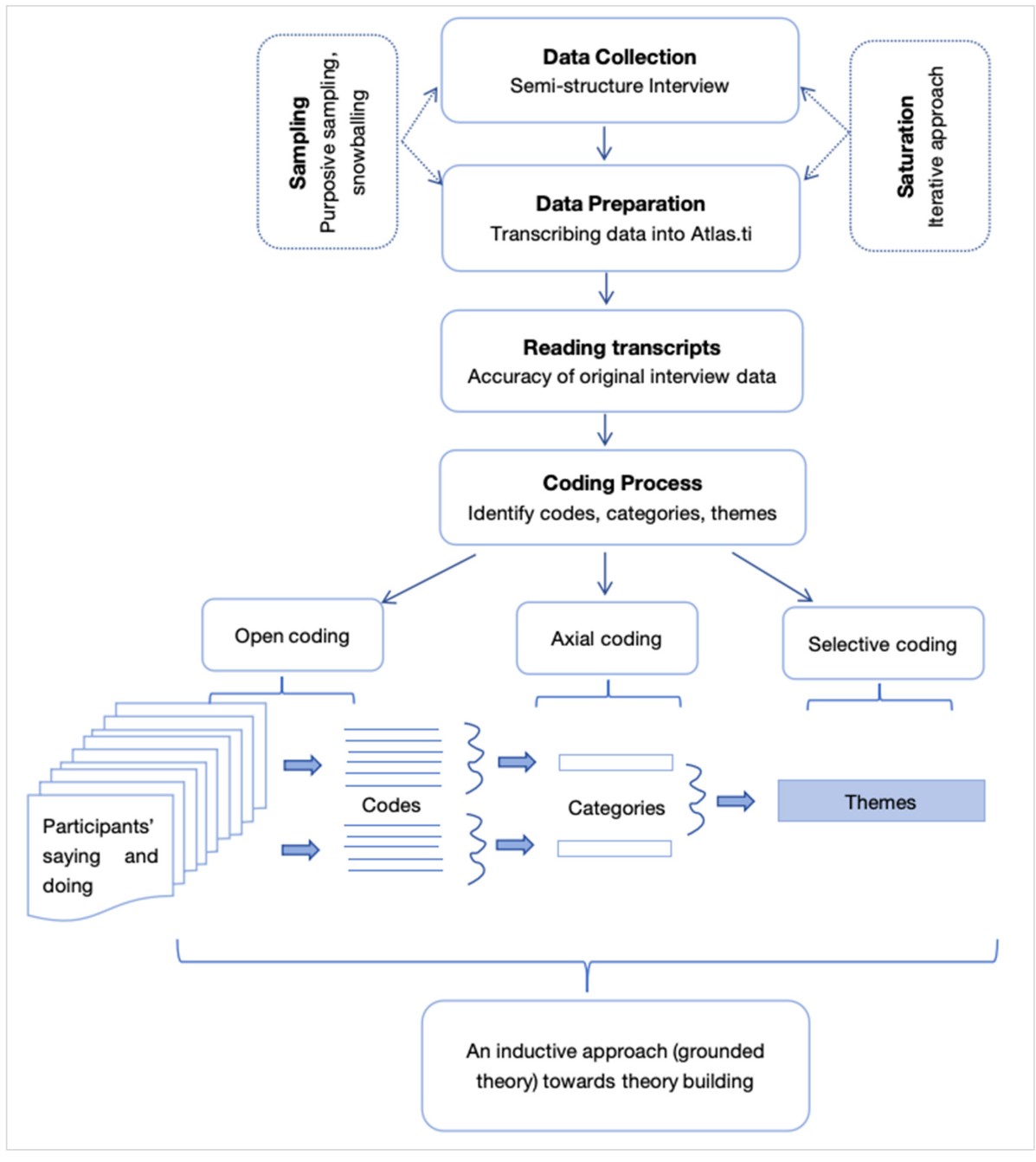

**Figure 1.** The data analysis process of this research.

To ensure the accuracy of our final themes and categories in reflecting the phenomena under investigation, we applied respondent validation by inviting participants to provide feedback for our results [111]. Additionally, to further enhance the validity and strengthen the credibility of our research outcomes, we invited a colleague to comment on our data analysis [112]. Finally, we read through and checked all documents, coding, and categories, extracted important information, and translated it into English. The last iteration of coding resulted in a final set of 94 codes. Figure 2 shows the structure of categories and codes; more examples of code details and analyses can be seen in Sheets S2–S4.

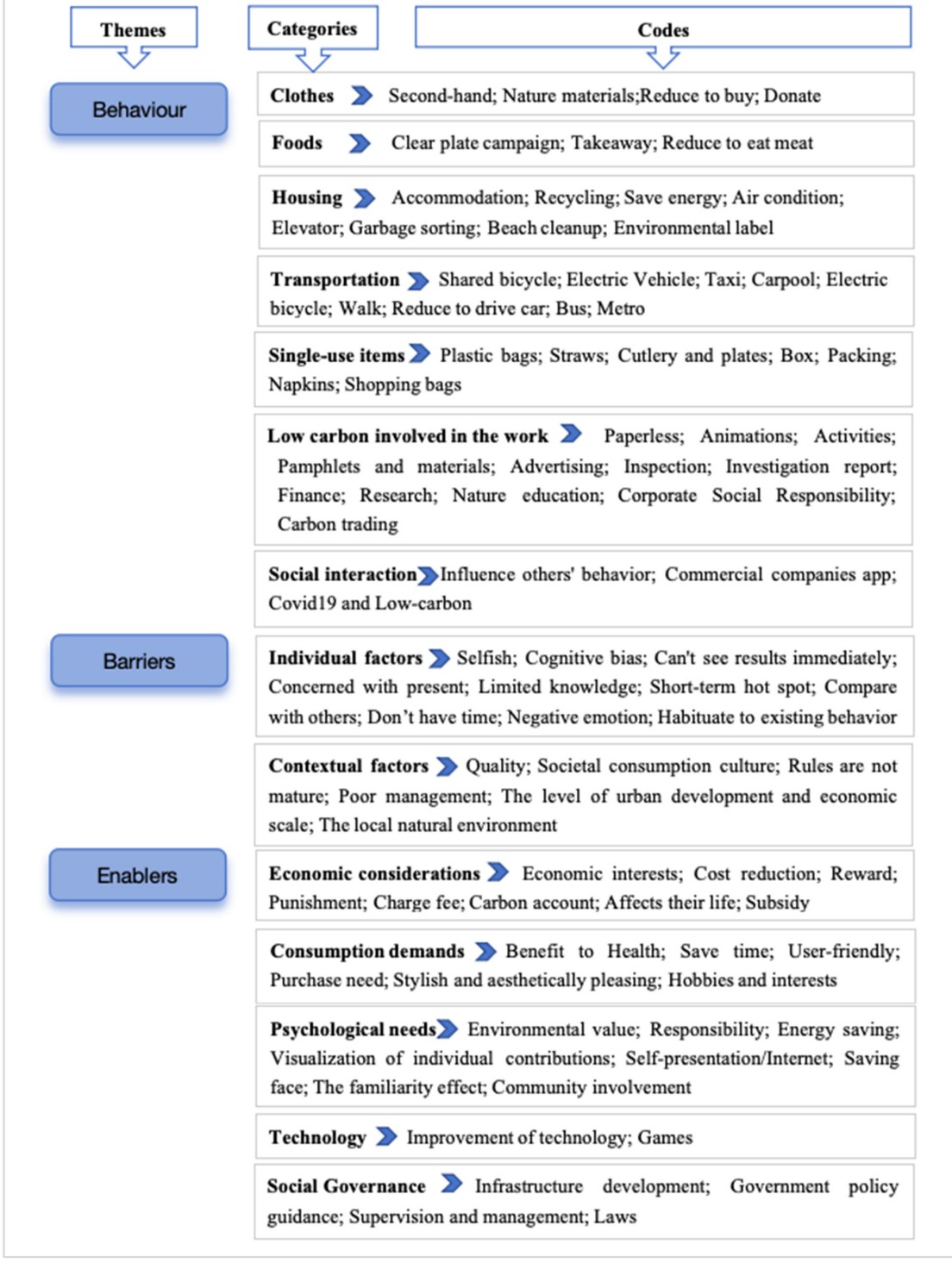

**Figure 2.** The structure of categories and codes.

## 3. Results

### 3.1. General Results

In terms of age, all of our participants were 27–42, belonging to the millennial generation. Among the interviewees, 18 held bachelor's degrees, 25 held master's degrees, 5 had Ph.D. degrees, and only 2 graduated from junior colleges. This represents that our article's results will show the daily life and consumption behavior of the well-educated millennial generation.

In our study of 50 participants, 24 were male, and 26 were female. We found that both male and female respondents had positive attitudes towards a low-carbon lifestyle. However, when it came to daily actions, women were more inclined to adopt low-carbon practices in household energy conservation because most of them are responsible for energy bill payments within their households. Additionally, married women played a key role in environmental education within their families, influencing their husbands and children through their low-carbon behaviors. One participant stated the following:

> In our new neighborhood, property management hasn't started, so we're not paying for water right now. We only pay the monthly electricity bill because it's managed by the energy company directly. I try to save water when I wash vegetables, do laundry, and take showers, and I also turn off lights promptly when leaving. However, my husband doesn't do so, I always remind him. P5

Our participants come from various backgrounds, including government (*n* = 4), traditional media (*n* = 16), corporate (*n* = 16), educational institutions (*n* = 3), residents (*n* = 9), and NGOs (*n* = 2). Figure 3 shows the specific work fields of participants. We found that some participants knew about a low-carbon lifestyle due to their work. Participants with prior experience in environmental protection are more willing to embrace low-carbon lifestyles because of their deeper understanding of environmental degradation and climate change impacts. In addition, female participants with low-carbon work experience or involvement in climate change action at work are fewer than male participants. Among the females who have related experience, most occupy execution roles rather than decision-making positions within their organizations.

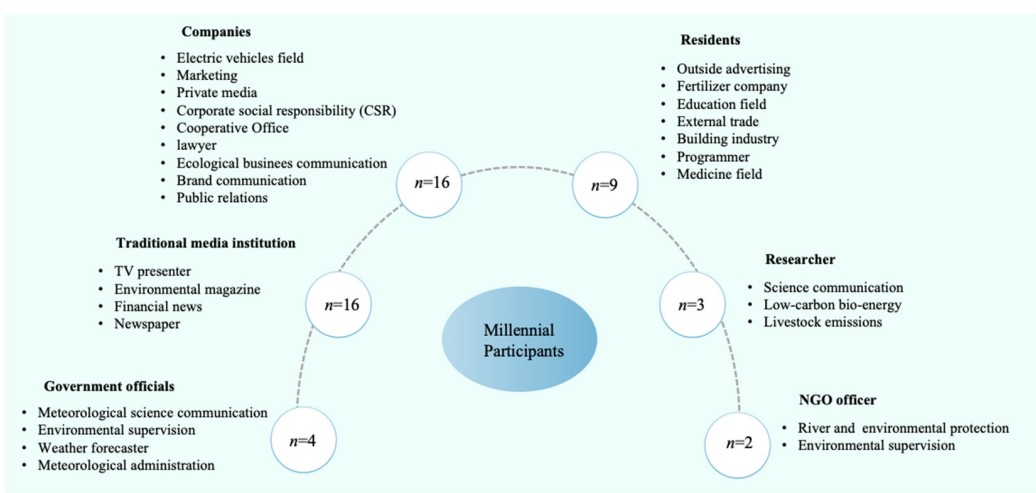

**Figure 3.** Work fields of participants.

In terms of residential areas, shown in Sheet S1 and Figure 4, 24 respondents live in China's first-tier cities. Among them, 18 participants live in Beijing, 4 participants are from Shanghai, and 2 of them are living in Guangzhou. Sixteen participants live in China's new first-tier cities (Nanjing, Suzhou, Hangzhou, Chengdu, Xi'an, Changsha, and Fuzhou), while ten respondents live in second or third-tier cities (Guiyang, Nanchang, Hohhot, Rizhao, Zhanjiang, and Changzhou). Our research indicates that city size impacts the millennials' lifestyle. For example, participants living in first-tier cities often rely on

subway commutes, and they also have a preference for ordering takeout. On the other hand, respondents living in second-tier or third-tier cities are more inclined to take buses or drive cars for commuting.

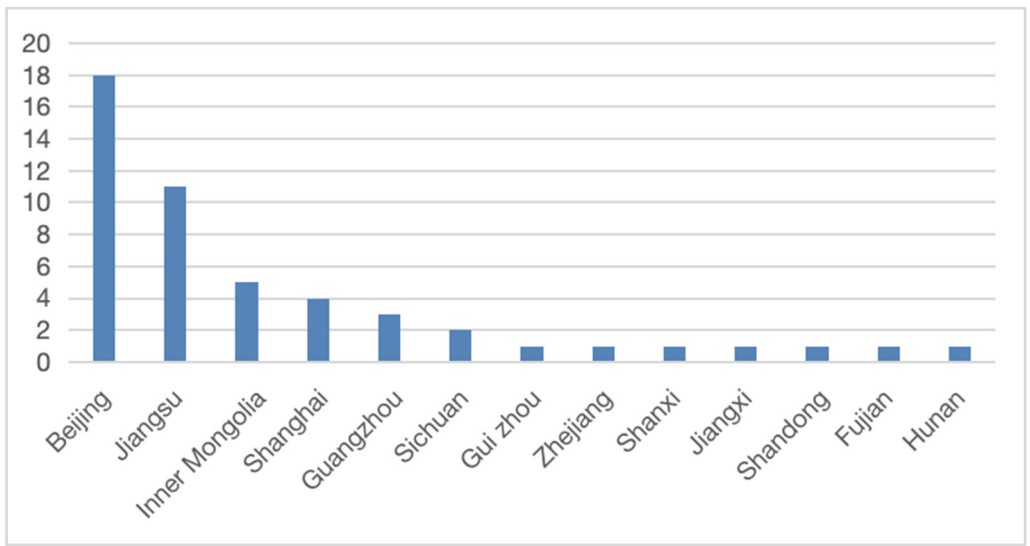

**Figure 4.** Areas of participants.

*3.2. Low-Carbon Behaviors of Millennials*

3.2.1. Low-Carbon Behaviors in Their Daily Life

Clothes. Participants practice low-carbon behaviors related to clothing by buying clothes with natural materials, reducing the buying of clothes, purchasing second-hand clothes, and donating or selling old clothes. Most of them prefer online shopping for new clothes, utilizing platforms such as Taobao or buying second-hand clothes through apps like Xianyu. They believe that buying less is key to reducing carbon emissions. Regarding recycling, the majority of respondents choose to either donate their old clothes at nearby second-hand clothing collection bins within their community or arrange for doorstep pickups with second-hand clothing sorting businesses. These businesses often offer compensation based on the weight of the clothes. Most respondents' motivation for dealing with second-hand clothing is not only to free up closet space but also to avoid wasting.

Foods. When it comes to low-carbon diets, respondents' initial reactions include reducing takeout orders, clean-plate campaigns, and consuming less meat. They believe that ordering takeout generates excessive single-use packaging, disposable utensils, and plastic bags. However, some respondents still would choose to takeout when they do not want to cook or need to save some time. Most respondents actively avoid wasting food and believe that the Clean-Plate Campaign is effective in China. This is because preventing food waste is a traditional value deeply ingrained in Chinese culture, and people have a higher awareness of this. During the Clean-Plate Campaign, some restaurants and cafeterias implemented rules to minimize food waste, and ordinary citizens participated in online monitoring. Furthermore, there is a growing trend among young adults to reduce meat consumption or transition to vegetarian diets, driven by health awareness and social environments.

Housing. Regarding housing, participants' low-carbon behaviors primarily include household energy conservation, waste recycling and utilization, and the purchase of eco-friendly home products. Firstly, the majority of participants exhibit household energy conservation habits, such as saving water and electricity and reducing the use of air conditioning and elevators. Among these practices, respondents demonstrate a stronger awareness of electricity conservation, as their monthly electricity bills exceed their water bills. Additionally, some participants like to reuse or renovate old items out of personal

interest, and others actively participate in beach clean-up activities due to the proximity of pollution sources to their homes. While many participants mentioned garbage sorting, most admitted to not practicing it in their daily lives; a lack of enforcement and management is the main reason. Thirdly, some participants expressed consideration for insulation effectiveness when purchasing a house and exhibited a preference for eco-friendly paints and other decorative products with eco-friendly labels. These choices are primarily motivated by concerns for energy efficiency and health.

Transportation. Compared to other low-carbon behaviors, participants show a higher level of awareness regarding low-carbon transportation. Many of them mentioned frequent use of public transportation, such as subways and buses, for their daily commute. Additionally, many participants regularly choose to use shared bicycles, particularly after disembarking from the subway or bus. Participants are conscious of the emissions associated with private cars and have opted for alternatives, such as taking taxis, carpooling, or purchasing electric vehicles. However, there are still some participants who continue to rely on gasoline cars for the commute because they purchased such vehicles early and do not want to buy new ones. Furthermore, some participants mentioned commuting via electric bicycles or walking to work regularly. Regardless of the chosen mode of transportation, most participants prioritize time and cost savings over environmental factors when making their commuting decisions.

Single-use items. Most participants mentioned that they try their best to reduce the use of plastic bags. Some participants stated that they use eco-friendly bags for shopping and avoid excessive packaging. Additionally, some participants mentioned that they actively reduce their use of disposable utensils, straws, and napkins while reusing disposable boxes. We found that the primary reason for respondents to reduce their use of plastic bags is due to plastic bag regulations, with many supermarkets now charging for plastic bags. Therefore, many people bring their own shopping bags.

Example quotes of participants can be seen in Table 2, which shows the above personal choices in terms of clothing, food, housing, transportation, and daily life about low-carbon behavior.

**Table 2.** Example quotes of participants' low-carbon behaviors.

| | Categories | Example of Quotes |
|---|---|---|
| | Clothes | Many communities now have individuals who collect clothes and offer doorstep pickups. It was convenient to gave them my clothes without the need for sorting, and they gave me a little money, usually ranging from 0.5 to 2 RMB per kilogram. P30<br>If there is a stylish coat with a fur collar, I would choose faux fur rather than fur sourced from animals. P34 |
| Section 3.2.1 | Foods | The Clean Plate Campaign in China is quite effective. Even among us adults, you see most people packing up their leftovers. I think this concept is not just about propaganda; it aligns with a traditional Chinese way of life. It's also about saving face. P29<br>I try to avoid ordering takeout as much as possible because it generates a lot of waste. P15 |
| | Housing | The situation of waste sorting is vary from city to city. In Suzhou, my community has a specific schedule for garbage collection: from 6:00 AM to 8:00 AM, and from 6:00 PM to 8:00 PM. During these times, two garbage bins are placed outside, and residents can dispose of their garbage. Dedicated individuals supervise the process, and they also help people for further sorting. We mainly separate the kitchen waste and other types of garbage. P19<br>In our community, we do have separate garbage bins for classification, but few people do so, including me. Most people are simply throwing the garbage into the bins without separation. P5 |

**Table 2.** *Cont.*

| Categories | | Example of Quotes |
|---|---|---|
| | Transportation | Shared bicycles have perfectly solved the last-mile problem because it takes 10 min to walk from the subway to my home, but I can get there in three minutes by riding a bicycle. I don't even have to buy a bicycle, and there are many shared bicycles at the subway station. However, a terrible thing is that the bicycles at the subway station are often placed in a disorderly manner. P23 |
| Section 3.2.2 | Low carbon behaviour involved in the work | I'm involved in green financing, which supports the financing of industries related to renewable energy, such as carbon-neutral initiatives and the electric vehicle sector. Currently, green financing also benefits from policy and financial support. We introduce our financial products to certain businesses, and if a company reduces its carbon emissions and meets the standards, their financing costs can be lowered. P26 <br> I participated in environmental monitoring reports previously. The provincial environmental protection department visits to some cities and companies then compiled the results, which were handed over to the local governments for follow-up. We then produced special reports based on these information. We visit the same places where they had been, and we assess how well these companies had implemented corrective measures. We then broadcast these findings on television. P7 <br> Last year, I cooperated with a research institution to Mount Everest to collect samples for a month and filmed a documentary about glaciers. Through this experience, I gained a deeper understanding of how climate change is closely related to our lives. P36 |
| Section 3.2.3 | Low-carbon behavior in Social interaction | I like to use Ant Forest because their concept is innovative, allowing users to convert their daily low-carbon lifestyle actions recorded through Alipay mobile payments into energy. Once a certain amount is accumulated, we can virtually plant a tree on the Alipay platform, and Ant Financial will also plant a real tree in the severely western desert areas of China. I believe participating in this activity is meaningful. My earliest awareness of low-carbon life came through Ant Forest. This approach, combining internet technology, finance, and low-carbon principles, makes it relatively easy for everyone to participate without any costs. P17 <br> Waste sorting in Shanghai was progressing well before the pandemic. However, the pandemic had a significant impact on waste sorting efforts, and very few people in my community continued to do so. People's priorities shifted during the pandemic as they had to address various immediate challenges in their lives. P10 |

(Note: More quotes of this part can be seen in Sheet S2).

### 3.2.2. Low Carbon Behavior Involved in the Work

Interviewees' low-carbon behaviors in the work include conserving energy in the office, engaging in low-carbon education, environmental monitoring and communication, carbon finance, etc. Some respondents mentioned that they make an effort to reduce paper usage and opt for double-sided printing, primarily due to their organization's requirements and to minimize waste.

In addition, some interviewees have work experience in environmental monitoring. Some of them work for the government, where they carry out environmental inspections. Others work in the media, investigating environmental pollution, producing reports, and making climate change-related promotional videos. There are also interviewees from NGOs who collaborate with the government and media to supervise polluting companies. These interviewees believe that their strong environmental awareness in daily life is due to their exposure to environmental degradation through their work. They also believed that low-carbon knowledge and personal behavior changes are mainly from their work.

Example quotes of interviewees' low-carbon behaviors in their work can be seen in Table 2.

### 3.2.3. Low-Carbon Behavior in Social Interaction

The participants' low-carbon behaviors also extend to social interactions, including influencing others' low-carbon actions, using Ant Forest, and behavior changes due to the

pandemic (COVID-19). Some participants mentioned that in their daily lives, they remind their family members to conserve energy, recommend environmentally friendly products, and purchase links to friends. Given their reliance on online shopping, entertainment, and social interactions, online product reviews and recommendations from popular social media influencers also play a significant role in shaping their purchasing choices. Additionally, participants reported that the environmentally friendly and low-carbon behaviors their children learn at school influence their own low-carbon behaviors and awareness. Furthermore, some participants mentioned Ant Forest, and most of them have been participating in the game for over three years because of its simple design, interactive rankings with friends, and easy integration into daily life. Ant Forest is an energy accumulation charity program initiated by Alipay, and as of April 2019, it had over 500 million users and had planted over 100 million trees in desert areas [113].

Furthermore, some participants reported significant changes in their lifestyles during the pandemic (COVID-19). Participants believe that during the pandemic, people were more focused on their personal lives and less inclined to consider low-carbon issues. While some individuals proactively reduced their carbon footprint, such as reducing shopping due to decreased personal income, others reported that they were in passive low-carbon behaviors, such as working from home, reducing takeout orders, and using public transportation less to maintain social distancing.

The above quotes from participants' low-carbon behaviors reflect the shift from the high-carbon lifestyle of the millennial generation, which requires not only an increase in individual low-carbon awareness and behavioral changes but is also influenced by personal interests, external pressures, social contexts, and policy support. The development of the internet has a significant impact on their shopping, dining, and travel behaviors. In addition, the quotes also demonstrate that low-carbon actions in the work fields are often connected to other individuals or organizations and are primarily achieved through collaboration with others. Our participants' work experiences related to low-carbon efforts range from raising environmental awareness among a broader audience to facilitating corporate transitions toward low-carbon practices, as well as supervising and rectifying high-pollution and high-emission enterprises.

*3.3. Barriers to a Wider Range of Low-Carbon Behavior*

Although many respondents maintain a positive attitude toward low-carbon living, their current low-carbon behaviors primarily stay in saving electricity, adopting low-carbon transportation, and reducing the use of plastic bags. Transitioning to a wider low-carbon lifestyle still has a long way to go. Therefore, we explore the barriers that influence the millennials' low-carbon behaviors.

3.3.1. Individual Factors

From a personal perspective, respondents believe that many people are primarily concerned with short-term personal satisfaction and lack deeper knowledge of low-carbon concepts. Most respondents pointed out that people tend to focus on the visible aspects of their lives or prioritize convenience, habits, saving time, and personal interests. They may not see their contributions to the environment or the impact of their actions in the short term. Furthermore, some participants reported limited knowledge of low-carbon products, and cognitive biases are also the reasons why people are reluctant to take action. For instance, participants mentioned that they were unclear about how to properly classify their waste despite being aware of the concept of waste sorting. This indicates the importance of climate change education, raising low-carbon awareness, and promoting more sustainable and low-carbon lifestyles.

> Some products are made of recycled materials, and some consumers may have cognitive bias, they think that these recyclable materials are extracted from trash. Therefore, some commercial advertisements may not specifically emphasize that

the products are made from recyclable materials, which can make consumers doubt the quality and safety of the products. P3

I believe that many people may feel that their small daily wasteful habits do not have a significant impact on the broader environment and climate change. Consequently, they may not consider changing their behavior. P5

The government and the media have been talking about carbon neutrality and climate change, but for ordinary people, it's just a concept. There is no clear understanding of what we need to do to support or participate in climate change mitigation. P41

### 3.3.2. Contextual Factors

As for external factors, participants reported that when engaging in low-carbon consumption, they consider the quality and safety of low-carbon products. Participants also mentioned that there is currently no established culture of low-carbon consumption, and the availability of low-carbon products on the market is limited, providing them with fewer choices. They also believe that limited media coverage about low-carbon lifestyles affects people's behavior. Furthermore, some participants indicated that their modes of transportation and the natural environment of their cities play a significant role. For example, in cities with hilly terrain, local residents are less likely to commute by bicycle. Moreover, most participants believe that bringing about change solely through individual voluntary actions, without rules and reasonable management, is not easy. However, some participants also believe that excessive constraints and mandates could lead to negative emotions and reluctance to act. This shows that guiding people toward low-carbon lifestyle changes requires balanced rules while considering individuals' freedom and well-being.

Some low-carbon products may have quality issues. For instance, certain degradable plastic bags or straws may not be very durable. P2

Due to space constraints, most media outlets do not cover climate change or low-carbon lifestyles on a daily basis. To some extent, this affects the scope of information that people receive. The level of professionalism in media coverage also influences people's interest in reading. P33

These barriers to adopting a more comprehensive low-carbon lifestyle indicate that achieving a broader transition toward a low-carbon lifestyle among millennials requires the cooperation of different stakeholders. For example, improving education and raising awareness about climate change through educational institutions and the media, encouraging businesses to transition toward low-carbon production, and having governments strengthen supervision and improve regulations. Sheet S3 shows additional example quotes of the barriers influencing participants' transition toward a low-carbon lifestyle.

### 3.4. Enablers to Switch to Low-Carbon Lifestyle

### 3.4.1. Satisfy Personal Needs

Economic considerations. Participants mentioned that people's interest in and attempts at low-carbon transformation are driven by their own economic interests. Some participants believe that when climate change disasters start affecting people's lives and causing more economic losses, there will be a stronger awareness of changing high-carbon lifestyles. However, by that time, it might be too late. Participants also indicated that if low-carbon products were more affordable and helped them save money, they would consider purchasing them. Conversely, if the prices of non-environmentally friendly products increased or if previously free services started charging fees, they would reduce their consumption and use. In addition, participants mentioned rewards and punishment mechanisms. Some participants believed that both material and psychological rewards, such as cash subsidies, reward points accumulated in carbon accounts for vacation benefits, or certificates of honor, could encourage low-carbon behaviors. However, others felt that rewards might be optional for some individuals, but punishment could be more effective.

Consumption demand. Participants reported that they would consider changing their consumption habits if low-carbon products could meet their daily consumption needs. They mentioned that their primary consideration would be whether they actually need the product rather than consuming it solely because it is low carbon. Some participants indicated that factors such as the product's potential health benefits, time-saving attributes, and ease of use would motivate their low-carbon consumption. Others mentioned that they are driven by the desire for personalized design and the fun aspect of products, as these help them showcase their individuality and aesthetics.

Psychological needs. Participants believe that certain personal values and social needs can drive them to change lifestyles to satisfy their inner needs, gain self-fulfillment, and obtain social acceptance. In terms of personal values, some participants mentioned that environmental values influence their behavior, such as reducing meat consumption for animal protection and organizing environmental protection projects in their work. Participants also reported that a sense of responsibility for the next generation motivates them to save energy and educate their children. Additionally, participants mentioned that if they could visualize their individual contributions to reducing carbon footprints, it would improve their confidence and motivation to change their lifestyles.

In addition, from a social perspective, some participants mentioned that they sometimes would consider the 'familiarity effect' or 'saving face' in their low-carbon actions. For example, when they see their colleagues turning off lights in the workplace or avoiding food wastage, they also follow. This helps them better integrate into their social circles. Furthermore, participants also mentioned that their self-presentation needs, particularly on social media, drive their low-carbon actions because they want to shape their social identity online.

### 3.4.2. Technology

Participants believe that technological advancements can drive people to change their lifestyles. In terms of consumption, they believe that technological progress in low-carbon products can lower production costs while improving product quality, meeting people's consumption needs, and increasing trust in these products. For example, many participants consider purchasing electric vehicles because of the maturity of the technology. In addition, participants also believe that the development of internet technology has transformed their lives, providing convenience and more choices in various aspects such as food delivery services, shared bicycles, online shopping, social media, virtual studio of TV programs, and live-streaming platforms. Therefore, they believe that future transformations toward a broader range of low-carbon lifestyles will also require technological innovation to guide low-carbon actions. However, some participants expressed concerns that while they enjoy the faster and more convenient services facilitated by technology, it could also lead to the emergence of new forms of waste, such as excessive packaging in takeout and online shopping. They believe it is necessary for the government to enhance relevant policies and further advance technology.

### 3.4.3. Social Governance

Participants believe that the government plays a crucial role in social governance, which includes low-carbon policy making, strengthening supervision and management, improving relevant laws, and enhancing urban low-carbon infrastructure. Participants reported that a broader transition to low-carbon lifestyles requires the government to establish reasonable policies and guidance, such as price controls and subsidies for new energy vehicles. Participants also mentioned that their awareness of low-carbon lifestyles improved due to the dual carbon goals impacting various industries, consequently affecting their lives.

In addition, participants mentioned that the government should strengthen supervision and management of waste and environmental pollution. They believe that if the government enhances management and improves relevant regulations, more people will

participate in waste sorting. Furthermore, participants also mentioned that the government needs to provide infrastructure support, which can influence their choice of transportation. For example, building more green pedestrian paths or bike lanes can encourage them to choose low-carbon transportation methods. Example quotes could be seen in Table 3.

**Table 3.** Example quotes of enablers to switch to low-carbon lifestyle.

| | Categories | Example of Quotes |
|---|---|---|
| Section 3.4.1 | Economic considerations | Economic incentives are quite practical, and the simplest form is economic subsidies. People are willing to participate; for instance, many people purchase new energy vehicles because of substantial government subsidies, which make the purchase price cheaper. P4<br>I have an electric vehicle, since I frequently business travel, using the electric vehicle instead of my previous gasoline car saves me a significant amount of money. There are numerous charging stations along the roads, and I can also avail of free battery swaps for recharging. P45 |
| | Consumption demand | If it's related to health, I tend to prioritize it. For example, when renovating a house, using eco-friendly adhesive that emits fewer harmful gases than regular adhesive or using water-based paint instead of solvent-based paint that emits fewer harmful fumes is not only environmentally friendly but also chosen primarily for the sake of one's own health. P3 |
| | Psychological needs | We need a specific data to gain confidence and hope, understanding that everyone's individual efforts, when accumulated, can contribute to mitigating climate change. This way, we will be more motivated to participate, as otherwise, many people may feel that our individual impact is insignificant. P2<br>Sometimes, I have a need for self-presentation. For example, I want to portray myself as an environmental enthusiast and refine my image on social networks. Building this personal brand may help me get more followers and more recognition from others. P28<br>We engage in environmental protection, such as organizing volunteer-led clean-up activities in Tibet or arranging charity events to assist pneumoconiosis patients. Our primary motivation stems from our values, even though the impact of these activities were limited. P48 |
| Section 3.4.2 | Technology | Compared to traditional cars, electric vehicles now have some advantages. Smart features of an electric vehicles can provide users with a better experience. In addition, as the quality of electric cars continues to improve and charging infrastructure expands, users' criticism towards electric vehicles has decreased. These improvements are mostly attributed to technology. P16<br>The improvement of technology in low-carbon communication, such as utilizing AI technology to simulate climate change scenarios and future potential risks, can enhance people's engagement and leave a more profound impression. P34 |
| Section 3.4.3 | Social governance | Policy making: I believe policy influence is significant. Influenced by national policies, now some big cities have implemented restrictions on driving, license plate issuance, and car purchase. For example, in Beijing, the government controls the ratio of issuing license plates for gasoline cars and new energy vehicles each year, making it easier for new energy vehicles to obtain licenses. Moreover, the government provides subsidies for new energy vehicles, which is a very important reason for people to purchase them. P8<br>Supervision and management: The reason why waste sorting is difficult to implement is because of the lack of supervision and reasonable rules. Unless the community is very strict, like in Shanghai, where each community has someone responsible, otherwise it is difficult for people to take action. P21<br>Infrastructure support. Some cities have bike lanes that are not very cyclist-friendly and are often encroached upon by private cars. Encouraging people to choose low-carbon transportation requires a better road infrastructure support. P5 |

(Note: more quotes on this part can be seen in Sheet S4).

The above quotes show the perspectives of the participants on the enablers of low-carbon behavioral change. The transition to a low-carbon society requires synergy among stakeholders, considering individuals' economic interests, consumption needs, and psychological needs. Reasonable policies and technological developments play a crucial role in encouraging millennials to participate in low-carbon transition. Figure 5 shows the theoretical model of the relationship between behavior, barriers, and enablers of millennials, as identified in our research.

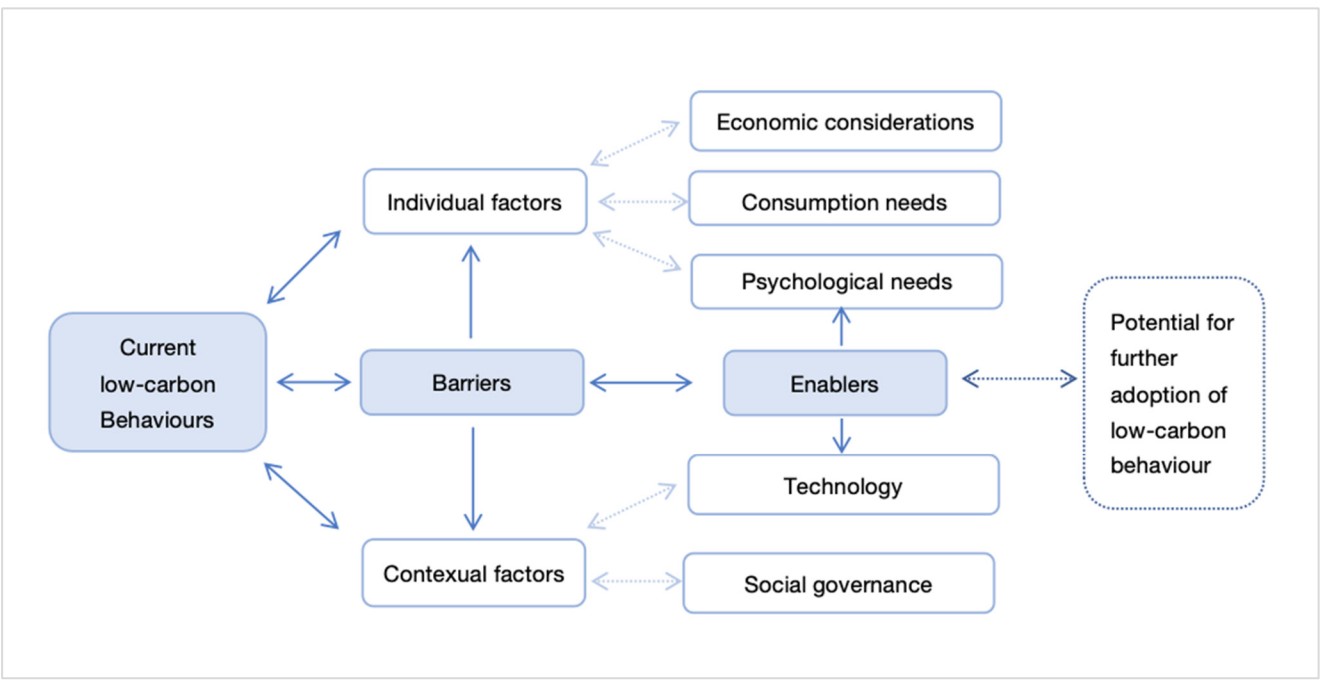

**Figure 5.** Theoretical model of the relationship between key themes in this research.

## 4. Discussion

This research provides a comprehensive overview of the low-carbon behaviors adopted by Chinese millennials in their daily lives based on our study samples. It explores the barriers and enablers that influence young adults in transitioning to a wider range of low-carbon lifestyles through in-depth interviews. Our research indicates that many Chinese millennials have positive attitudes towards a low-carbon lifestyle. Currently, their low-carbon behaviors mainly focus on saving electricity, using shared bicycles, reducing plastic bag usage, minimizing paper usage in the workplace, and taking public transportation for commuting. The main barriers for millennials in transitioning to a wider low-carbon lifestyle include the difficulty of seeing immediate contributions and impacts on the environment, more concern for the current situation rather than climate change in the future, and a lack of deeper knowledge about low-carbon behaviors. Millennials would change their lifestyle mainly for economic benefits, consumption needs, and social interaction. Reasonable policies, technical innovations, and cooperation from stakeholders in different fields will encourage more people to join in low-carbon action. Our findings provide the following three core insights.

Firstly, our study samples indicate that Chinese millennials have a positive attitude towards transitioning to a low-carbon lifestyle. Women demonstrate a stronger willingness to adopt low-carbon behaviors in their daily household activities compared to men. However, their involvement in governance in the context of transitioning to a low-carbon society is limited. Our research indicates that married women play a crucial role in environmental education within their families, and their low-carbon behaviors influence their husbands and children. This is primarily because modern urban women in the Chinese millennial generation have received a good education, leading to a higher environmental awareness. Over the past 40 years of China's reform and opening up, the proportion of millennial women receiving higher education has gradually increased. The percentage of female graduate students in China reached 50.9% in 2020, surpassing that of their male students [114]. However, our case study shows that, in the context of transitioning to a low-carbon society, female participants with experience in climate change or low-carbon activities are fewer than male participants. Within organizational management, women often assume execution roles in climate action rather than decision-making positions. Based

on previous research, climate change and related policies will involve issues of gender equality, particularly in developing countries [115–119]. Encouraging more women to participate in the social innovation process will effectively contribute to mitigating climate change [120]. Some millennials are already or will soon become part of organizational management; they will play a key role in climate action over the next 30 years. Jiang et al. researched Chinese companies in manufacturing industries with serious carbon emissions and found that executives' low-carbon attitudes indirectly affect the performance of carbon emission reduction in their companies [121]. Martínez et al. [122] analyze the relationship between gender on company boards and $CO_2$ emissions in European companies, and they find that a higher proportion of female members on corporate boards correlates with lower carbon emissions. Altunbas et al. [123] observed that an increase of 1% in female managers is associated with a 0.5% decrease in $CO_2$ emissions in the organizations. Therefore, our research suggests that the transformation towards a low-carbon society needs to consider gender equality. It is essential to amplify the voices of women and provide them with more opportunities in the decision-making process. Additionally, this requires the improvement of more female managers' low-carbon transition awareness and active participation in more low-carbon activities at work.

Secondly, technological innovation and progress play a crucial role in the transition to a low-carbon lifestyle among the millennials. However, they also come with new forms of resource waste, so making sensible policy formulation is the key. In the millennials' current life, whether it is food delivery services, bike-sharing, online shopping, social media platforms, or payment platforms, all rely on the development of mobile internet technology. Technological innovation could also enhance the quality of low-carbon products, improve household energy efficiency, and reduce people's living costs. However, the millennials, while enjoying the convenience of rapid online consumption and services, also leading to new forms of resource waste and environmental issues. For instance, excessive packaging generated by takeout and online shopping, the widespread use of disposable utensils, and the damage and idle occupancy of public spaces by shared bicycles. A study conducted by Echegaray et al. [124] in Brazil indicates that while most people have a positive awareness of recycling electronic appliances, only a few of them actually implemented e-waste recycling practices. The tension between economic development and environmental protection is a common challenge faced by contemporary society [125]. Ecomodernism believes that technological development can protect nature and improve human well-being, but currently, there is limited explanation of how to protect the substantive freedom of vulnerable populations during the climate change crisis [126]. The "Jevons Paradox", proposed by the British economist William Jevons, suggests that although modern technology can enhance resource efficiency, the decrease in usage costs leads to an increase in demand [127]. Consequently, this increase in demand results in higher, rather than lower, resource consumption [128]. In other words, relying solely on technological advancement cannot help humanity reduce the overall consumption of resources and energy [129,130]. Our research believes that in the transition to a low-carbon society, technological innovation and application need reasonable policy guidance and effective supervision to prevent new forms of waste [131]. People's low-carbon awareness should be improved through education and media by disseminating more details of low-carbon knowledge and guiding individuals on how to take action in their daily lives. Collective action will contribute to ensuring that technological progress genuinely serves the achievement of sustainable development in terms of resources, environment, and society.

Thirdly, personal economic interests are the main driver for millennials to switch to a low-carbon lifestyle. Reasonable policies can guide them toward a low-carbon lifestyle, but this requires synergies among stakeholders. Our research has found that millennials, whether in the purchase of electric vehicles, reducing the use of plastic bags, or increasing low-carbon activities at work, have been influenced by relevant policies. These policies include incentives, price adjustments, supervision, and management. The low-carbon transformation not only relates to the economy and the environment but also involves

social interactions and policy-making, which requires multidimensional synergy [132–134]. Improving public low-carbon awareness and behavior is crucial to low-carbon city transformation [135]. Most millennials have transitioned from formal school education to broader social learning. We found that many of them learn about low-carbon practices through their work, as the dual carbon goals and related policies in China have impacted various industries, further affecting their lives, so the improvement of low-carbon knowledge training is necessary in related industry. Our research suggests that universities, enterprises, and the government should strengthen cooperation to enhance the quality of low-carbon products and services, align with the life needs of millennials, and reduce the prices of low-carbon products. Moreover, efforts should be made to offer more easily understandable and actionable low-carbon guidance. These will encourage more millennials to switch to a low-carbon lifestyle. In addition, due to the impact of COVID-19, an increasing number of millennials are focusing on their health, leading to a shift in their low-carbon lifestyle. Our research believes that a wider range of low-carbon society transitions requires the establishment of a coordinated and comprehensive social governance that is credible, flexible, and reflective. This should incorporate different backgrounds, cultures, values, and anticipate future changes to continuously revise policies [136]. Additionally, it should consider people's well-being and happiness.

## 5. Contributions, Limitations, and Future Research

Previous research about China's low-carbon transition has mainly focused on exploring the concepts and policies related to low-carbon cities, analyzing the models of low-carbon pilot cities, and calculating household carbon emissions. Our research will have several contributions. In terms of theoretical contributions, first, we use in-depth interviews to focus on the attitudes, behaviors, and past low-carbon activities of the Chinese millennials. Although we have 50 sample interviewees, compared to questionnaire methods with larger sample sizes that only provide approximate written information, interview methods prioritize depth and understanding of the experiences and sentiments of the respondents. This emphasis enhances the reliability of the research's authenticity. Through interviews and in-depth conversations with participants, we also learned about their attitudes toward their peers' lifestyles. By deeply communicating and discussing with participants according to our theme, the discussion of low-carbon lifestyles extends beyond the 50 participants. All of our participants are from different industry fields in China; many of them are related experts and journalists who are knowledgeable about related social issues and low-carbon ideas. Our research will help scholars get a better understanding of people's daily low-carbon choices and practices, which provides a foundation for future scholars to further explore individual low-carbon behavioral preferences, personal carbon footprint calculations, public participation in low-carbon societal transitions, etc. Secondly, by showing real-life cases from the respondents, we analyzed the barriers and driving factors of low-carbon actions from psychological, sociological, and economic perspectives. This multidisciplinary approach will assist scholars in exploring the influencing factors of low-carbon consumption behaviors. Thirdly, our study highlights the significant role played by women in addressing climate change and the potential problems related to technology participation in climate action. This offers new perspectives for future scholars to explore climate action, helping deepen the understanding of the roles and challenges faced by different social groups in low-carbon transitions. Furthermore, in terms of practical contributions, understanding the lifestyles, driving enablers, and barriers faced by the millennial generation will help stakeholders consider the actual needs of this generation when formulating low-carbon policies and providing related services. This understanding will guide millennials to adopt low-carbon lifestyles and actively participate in collective climate action. During our in-depth interviews, our research also helped our participants rethink their low-carbon behaviors and related work. This encouraged them to contribute more to the transformation of low-carbon society in their respective fields.

This study has some limitations. Firstly, our research focuses solely on China's millennial generation. Therefore, the findings might not be directly applicable to low-carbon societal transitions in other countries with different policies, cultural backgrounds, and levels of economic development, although our case studies could serve as a reference. Secondly, this study may face criticism for the limitation of sample size in qualitative research. We chose a semi-structured interview due to its depth and flexibility in conversation, as well as the lack of research focusing on the millennials' low-carbon behavior in China. While our study provides in-depth insights and rich data based on our sample, addressing gaps in the literature, our research results may not capture the entirety of millennials' lifestyles in China. For example, our study is based on participants from different cities, most of whom reside in first-tier or capital cities in China. Due to differences and imbalances in economic development, education, natural conditions, and cultural diversity among cities in China's eastern, middle, and western regions [137], our research results might not be universally applicable to other cities in China. Thirdly, with over half of our participants holding master's degrees or higher, our results only reflect the highly educated Chinese millennials' daily life and consumption behavior.

For future research, we suggest conducting studies focused on specific cities. These studies could integrate local customs and urban development characteristics, explore the low-carbon lifestyles of local residents, and compare consumption behaviors across different genders and generations. Additionally, we recommend future research to further explore women's role in climate change action in developing countries and investigate the relationship between female leadership and carbon emissions in China. We also encourage scholars to explore other research methods to discuss people's low-carbon lifestyle and behavior, such as combining interviews with panel data analysis or creating workshops as a qualitative research tool to observe the dialogs between participants from different fields and generations.

## 6. Conclusions

Climate change issues exhibit temporal lag, the misalignment between perpetrators and victims, and the non-renewability of climate resources. Different regions and populations have varying demands for greenhouse gas emissions, mitigation, and adaptive capacities, which not only involve environmental concerns but also intersect with social development, fairness, justice, human rights, and sustainable development issues. This study focuses on the millennials' low-carbon behaviors in China, who are the mainstay of the current labor market and will be the decision-makers and managers in the future for climate change action. They have significant potential to participate in collective climate action. This study uses in-depth interviews to explore the millennials' low-carbon behaviors, the barriers, and enablers for a wider range of low-carbon actions, which enhances the understanding of low-carbon lifestyles in China. We summarize and analyze the respondents' low-carbon behaviors in their lives, work, and social interactions, offering a holistic view beyond the focus on household-level carbon emissions often found in previous research. Our research provides a foundation for future studies on individual preferences, carbon footprint calculations, and public engagement in low-carbon transitions. From practical contributions, this research will help policymakers in formulating more rational incentive policies. It can also help business managers better comprehend consumer demands and find a balance between economic interests and emissions reduction, facilitating their own low-carbon transformations.

Based on our study samples, there is an indication that Chinese millennials have a positive attitude towards transitioning to a low-carbon lifestyle. Current low-carbon behaviors mainly focus on saving electricity, using bike-sharing services, reducing plastic bag usage, and reducing paper in the workplace. Women in households show a higher willingness for low-carbon actions compared to men. However, women who have experience in climate change or low-carbon activities in the workplace are fewer than males. Most women play an execution role in climate change actions rather than a decision-making position. Further,

millennials' low-carbon life transition is accompanied by technological innovation and progress. However, this progress would bring some new forms of resource waste, and reasonable policy-making and management will be the key. The difficulty in immediately perceiving their contribution and impact on the environment, coupled with a lack of deeper knowledge about low-carbon lifestyles, are the main barriers for millennials to switch to low-carbon action. Personal economic interests and the satisfaction of their consumption needs will drive millennials to reduce carbon emissions in their daily lives, but it requires reasonable policy-making and synergy among various stakeholders. As most millennials have transitioned from formal school education to broader social learning, many of them acquire low-carbon knowledge through their work. Therefore, enhancing low-carbon knowledge training in related industries is necessary. Technological advancements could also drive millennials to join in low-carbon actions by reducing the cost of low-carbon products and providing more easily understandable and actionable guidance.

**Supplementary Materials:** The following supporting information can be downloaded at: https://www.mdpi.com/article/10.3390/smartcities7040080/s1, Sheet S1: Area of participates living in; Sheet S2: Example quotes of participants' low-carbon behaviors; Sheet S3: The barriers influencing participants' transition toward a low-carbon lifestyle; Sheet S4: Enablers to switch to low-carbon lifestyle; Sheet S5: Interview Questions.

**Author Contributions:** Research design: Y.W., P.M. and T.K.; interview and data analysis: Y.W.; writing—original draft: Y.W.; writing—review: P.M. and T.K.; supervision: P.M. and T.K. All authors have read and agreed to the published version of the manuscript.

**Funding:** Yan Wu is funded by the China Scholarship Council (CSC) with grant No. 202008320395. The funder had no role in research design, data collection and analysis, decision to publish, or preparation of the manuscript.

**Institutional Review Board Statement:** This project has been reviewed and approved by the Faculty Niet-WMO Verplicht Research Ethics Committee, Maastricht University. Reference Number: FHML-REC/2023/090. Approval date: 23 October 2023.

**Informed Consent Statement:** Presented at the end of the article.

**Data Availability Statement:** The original contributions presented in the study are included in the article/Supplementary Materials, further inquiries can be directed to the corresponding authors.

**Acknowledgments:** We want to thank all participants for joining the interviews.

**Conflicts of Interest:** The authors declare no conflicts of interest.

## Appendix A

### Interview Invitation Letter

Dear Sir/Madam,
My name is Yan Wu, I am a PhD researcher at Maastricht University, supervised by Prof. Dr. Pim Martens and Prof. Dr. Thomas Krafft, currently doing my research project, 'Public awareness on low-carbon city transformation in China'. I'm funded by China Scholarship Council (CSC) with grant No. 202008320395.
Responding to global climate change is not only a national responsibility but also an individual responsibility. This interview is part of the research project "public awareness on low-carbon city transformation". In order to realize a low-carbon future, we need to understand the public's low-carbon awareness and behavior. We aim to learn about factors affecting public awareness towards low-carbon city transformation.
This interview is being used for research purposes only, not for any commercial purposes. All the answers will just represent individual opinions instead of your institution. To protect the privacy of the interviewees, this study will establish a numbering system for each interviewee, using a number instead of the name in the transcript. Thank you for accepting our interview.

## Research Questions

1. Age, gender, location, highest level of education, job occupation.
2. Current role in your organization. Describe your past environmental work experience.
3. What do you know about low-carbon lifestyle and low-carbon economy?
4. How do you know these?
5. How do you feel about the shift to a low-carbon economy and low-carbon life?
6. What type of transportation do you prefer to choose for short trips and long trips, and why?
7. What factors do you think affect people's awareness about low-carbon (why or for example)?
8. Do you buy low-carbon products in daily life? Why?
9. Do you think many people know that low-carbon will contribute to climate change and our environment, but sometimes it is difficult for them to change behavior?
10. What do you think can interest people to join in low-carbon activities and turn to low-carbon life and low-carbon consumption?

## Informed Consent Statement

1. I have read the interview invitation letter and understand that my role is interviewee in this interview.
2. I voluntarily agree to participate in the interview and know that I can stop the interview at any time.
3. I know I have the right not to answer or skip questions if I don't want to answer.
4. I agree that this study could use my responses anonymously as research data and be published in journals.
5. I allow the interviewers to take notes and record during the interview.
6. I agree that this research team may use my interview for scientific research on the project "Public Awareness of Low-Carbon City Transformation".
7. I have read and understood the points and statements of this consent form.
8. I have had the opportunity to ask questions about this interview.
9. I confirm and accept the interview.

_______________________          _______________________
Participants                               Date

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
