# Peer review of "Transitioning to a Low-Carbon Lifestyle? An Exploration of Millennials’ Low-Carbon Behavior—A Case Study in China"

_smartcities, doi:10.3390/smartcities7040080_

Round 1

Reviewer 1 Report (Previous Reviewer 1)

Comments and Suggestions for Authors

Reviewer report – Transitioning to a Low-Carbon Lifestyle? An Exploration of Millennials’ Low-Carbon Behavior – A Case Study in China

By Yan Wu, Pim Martens, Thomas Krafft

To better understand low-carbon lifestyles in China, this study used in-depth interviews to investigate the low-carbon behaviors of millennials as well as the barriers and enablers for a wider range of low-carbon actions.

The authors, compared to the last article, made improvements but mostly explained their approach.  Even if the interview samples size was too small, the results were nevertheless validated because a justification for the small samples was given.

Within any research area, different participants can have diverse opinions, as is the case here between reviewers and authors. The significantly high proportion of studies utilizing multiples of ten as their sample is the most important finding from the research literature. [https://doi.org/10.17169/fqs-11.3.1428 ]

Qualitative samples must be large enough to assure that most or all the perceptions that might be important are uncovered, but at the same time, if the sample is too large, the data becomes repetitive and, eventually, superfluous. This is the concept of data saturation invoked by the authors.

While saturation is a convincing concept, it has several practical weaknesses. The factors that can influence a qualitative sample size and, therefore, saturation in qualitative studies are the heterogeneity of the population, the number of selection criteria, multiple samples within one study, the types of data collection methods used, the budget and resources available, the scope of the study, the study design, and the information reported about the experiences of others, which is called shadowed data.

What is clear from this exposition is that there are issues prevalent in the article that are not wholly congruent with the principles of qualitative research. One individual, for instance, cannot be considered a sample. However, they might be considered to validate the other samples' results if their opinions align with the age and socio-professional demographic of the sample they belong to.  A point that the paper highlights is to avoid the requirement for a larger sample size and several frequently unreliable statistical tests.

The results of the research typically correspond with worldwide trends, such as the focus on environmental issues that are antagonistic to individual economic interests and the transition to a low-carbon lifestyle among millennials, which coincides with progress in technology.  Additionally, it can aid in business managers' low-carbon transitions by helping them better understand customer demands and strike a balance between financial interests and emissions reduction.

Because the manuscript provides information that readers of the journal will find interesting, it should be published. The results are relevant, and this analysis is an acceptable method.

Best regards,

The reviewer.

Author Response

Dear Reviewer,

Many thanks for your comments, please check the documents attached.

Kind regards,

Authors

Reviewer 2 Report (Previous Reviewer 2)

Comments and Suggestions for Authors

Due to the very extensive additions and revisions, the aim of the paper, the justification of the methods, and the implications are now very clear. I would like to commend the authors for their substantial contributions and revisions. In particular, I find the section examining the implications of gender in relation to decarbonized lifestyles to be a highly valuable study.

In my opinion, the paper is ready for publication following English editing. However, if you are still open to further improvements, I would recommend considering the following points:

Around Line 122, the paper mentions that the importance of the study is tied to the fact that the target population, millennials, are more likely to be in management positions. This is a crucial point. It would be beneficial if the results and discussion section also addressed how the findings and ideas of this paper might influence the rest of society when millennials are in management roles.

Lines 153-159 contain a description of the characteristics of Chinese millennials by Ata, which has little direct relevance to the validity of the method. This section might be better suited to Section 1, elsewhere in Section 2, or in Section 3.

3.2.1 Low-Carbon Behaviours in Daily Life

In this section, the subjects' views on clothes, food, housing, transportation, and single-use items are briefly presented. The analysis is clear and interesting. To enhance it further, it would be useful to show how these consumption areas relate to China's economic growth and GHG emissions or carbon footprint in urban areas. Including data from previous studies would provide a more convincing explanation of the subjects' views on these objectively important areas.

3.2.2 Low-Carbon Behaviour at Work

3.2.3 Low-Carbon Behaviour in Social Interaction

Similarly, these sections would benefit from a brief discussion on how the importance of emissions in workplace activities and social interactions is understood in China, based on existing research and other sources. This would help readers better appreciate the significance of these two sections.

Regarding the sample size mentioned in the Limitations section, I don't think it is the main issue. Rather, the study's use of the term "Chinese millennials" could be problematic, given that the subjects were selected through personal networks and snowball methods in a limited number of cities. It would be more accurate to specify that the study represents "millennials working in cities in the 00 region of China, in the 00 sector." This specificity would highlight the uniqueness of the study, rather than implying it represents all Chinese millennials.

Considering this, future research directions could include planning comparative studies on gender-focused differences between men and women and generations, or between men and women and the employment sector. Additionally, analyzing how new knowledge and practices emerge from dialogues between people from different fields and generations would be valuable.

Comments on the Quality of English Language

A thorough review of the English language and editing is strongly recommended. There are numerous errors in singular and plural forms. Additionally, conjunctions such as "therefore" were sometimes misused, with the preceding and following sentences not connected causally.

Author Response

Dear Reviewer,

Many thanks for your comments, please check the documents attached.

Kind regards,

Authors

Reviewer 3 Report (Previous Reviewer 4)

Comments and Suggestions for Authors

The abstract should include more information on methodology such as (i) sample dimension and type of quality research, and (ii) direct conclusion.

Using 50 samples to draw conclusions about "Transitioning to a Low-Carbon Lifestyle considering Millennials’ Low-Carbon Behavior in Case Study in China" is unacceptable (Fig. 3 is a perfect prove). This sample dimension might be used to study a small municipality of China, but not a big city or entire China.

Statistical methods were not used to accept or reject hypothesis which might be required for this sort of research.

Author Response

Dear Reviewer,

Many thanks for your comments, please check the documents attached.

Kind regards,

Authors

Reviewer 4 Report (Previous Reviewer 5)

Comments and Suggestions for Authors

1. This paper involves a large number of human subjects, even if it is an interview experiment, ethical exemptions cannot be allowed. Authors are required to provide ethics committee permission.

2. The depth of insight provided by semi-structured interviews is insufficient. The methodological contribution of this paper needs to be strengthened.

3. The sample size is relatively small (50 participants) and may not be fully representative of the low-carbon behavior of all Chinese millennials.

4. The main conclusion of this paper is oversimplified.

Comments on the Quality of English Language

I think it could have been revised better.

Author Response

Dear Reviewer,

Many thanks for your comments, please check the documents attached.

Kind regards,

Authors

Round 2

Reviewer 3 Report (Previous Reviewer 4)

Comments and Suggestions for Authors

To publish this work the authors must redefined the manuscript scope by limiting it to Beijing (18 sample) and Jiangsu (10 sample). Data/information on others regions must be removed due to data scarcity.

Therefore, the new title must replace China with Beijing and Jiangsu, and would be:

Transitioning to a Low-Carbon Lifestyle? An Exploration of Millennials’ Low-Carbon Behavior –A Case Study in BEIJING and JIANGSU.

There is no way to published it without the above mentioned change.

Note: Science must be rigour and responsible, so that the current version is unacceptable.

Author Response

Dear Reviewer,

Many thanks for your comments, please check the documents attached.

Kind regards,

Authors

Reviewer 4 Report (Previous Reviewer 5)

Comments and Suggestions for Authors

The authors have responded the reviewers' questions and revised the manuscript properly based on the reviewers' comments. Therefore, I recommend this version can be considered to publish.

Author Response

Dear reviewer,

Thank you for your comments and for reviewing our article.

Kind regards,

Authors

This manuscript is a resubmission of an earlier submission. The following is a list of the peer review reports and author responses from that submission.

Round 1

Reviewer 1 Report

Comments and Suggestions for Authors

Reviewer report – Transitioning to a Low-Carbon Lifestyle? An Exploration of Millennials’ Low-Carbon Behavior – A Case Study in China

By Yan Wu, Pim Martens, Thomas Krafft

I had previously reviewed this article once. The article uses semi-structured interviews to address an interesting subject. If the study conforms to the fundamentals of mathematical statistics, it offers excellent research potential. I also highlighted the strengths of the authors' study in the initial review. I must, however, draw attention to the major observations that I previously mentioned. These major observations are brought up by the non-rigorous statistical method.

The results could have been justified even if the interview sample size was too small if a detailed explanation of the shortcomings had been provided. The explanation assumed that statistical principles were not used and that future research directions were sought. It basically lays the foundations of a model for a research topic.

Major observations:

1.     The sample is insufficient, especially considering that it includes a diversity of cities and levels of education. As a result, it becomes impossible to correlate the categories with the answers provided by the interviewees. Any correlation is invalid since there is not enough data to substantiate the conclusions of the article scientifically.

2.     Statistical characteristics of the survey have to be presented, namely, the margin of error (which indicates how much error exists around a measurement), a confidence interval (which indicates how reliable a measure is), and the standard deviation (which describes how much the replies differ from person to person and from the average number).

3.     Graphs are absent from the section on data analysis and presentation. These would have been an interesting method to present the results, but this is unfeasible because of the tiny sample size.

4.     Include a detailed explanation of the data analysis method for increased methodological clarity.

The article's essential premise is interesting, but it still requires improvement to be considered a scientific article; as it remains, it is just a very good concept for a fascinating work of research.

It is my opinion that authors must be motivated to enhance the sample size and submit the paper again. When I first looked at it, I assumed that the sample was merely an excerpt from a larger sample. It seems not, and the writers wouldn't have enough time to expand the survey sample in a short amount of time. Thus, I am forced to turn down an article that sounds very promising but has no mathematical basis. 

Best wishes,

The reviewer

Reviewer 2 Report

Comments and Suggestions for Authors

This topic has the potential to be very insightful, exploring the perspectives of Chinese millennials on decarbonized lifestyles. However, some aspects of the research methodology warrant closer examination.  Specifically, areas like sampling, interview analysis, and how those findings translate into discussions and conclusions would benefit from further refinement.

The study employs snowball sampling to recruit participants, focusing on millennials in China. While snowball sampling can be useful for reaching specific populations, it may not be ideal for achieving a diverse sample representative of all Chinese millennials.  This is because snowball sampling tends to recruit individuals within existing networks, potentially missing out on millennials from different geographic regions, socioeconomic backgrounds, or occupations. To strengthen the generalizability of the findings, considering a sampling method that allows for more balanced representation across these demographics would be beneficial. Additionally, with this sampling approach, labeling the study participants solely as 'Chinese millennials' in the Discussion section may not be fully justified. A more nuanced description acknowledging the potential limitations in representativeness due to the sampling method would be advisable.

While in-depth interviews are a valuable tool, the current analysis approach (categorizing responses into pre-defined themes in sections 3.2-3.4) appears to be a basic thematic analysis that doesn't fully utilize the richness of the data. This approach primarily summarizes existing categories rather than exploring new insights or uncovering relationships between themes.

A more nuanced analysis could explore several aspects:

Novelty: Identify themes that emerged from the interviews themselves, going beyond pre-conceived categories.

Participant Differences: Analyze how responses vary by factors like gender, occupation, and income.

Interrelationships: Explore connections between categories like "Barriers" and "Enablers" for specific behaviors mentioned by participants.

Lines 656-658 highlight an important point. Interviews are about co-constructing knowledge. Analyzing the questions asked, responses received, and how both interviewer and interviewee perspectives shifted during the conversations could reveal valuable insights even with a limited sample size.

The Discussion section (Section 4) raises several insightful points, particularly regarding gender and climate change as well as the potential downsides of technological solutions in ecological modernization. However, these themes seem absent from the analysis presented in Section 3.  For a stronger connection between data and conclusions, it would be beneficial to ensure that the analysis in Section 3 reflects the topics subsequently discussed in the Discussion section.

The study explores a fascinating topic with a well-structured thematic framework. The Discussion section raises insightful points that resonate with us. However, to fully capitalize on the research potential, strengthening the research design and analysis methodology would be valuable. This could involve addressing areas like sampling strategy and interview analysis techniques.

Reviewer 3 Report

Comments and Suggestions for Authors

Having been given the opportunity to review the revised manuscript titled "Transitioning to a Low-Carbon Lifestyle? An Exploration of Millennials’ Low-Carbon Behavior – A Case Study in China" for a second time, I have carefully examined the changes made in response to my previous comments. I am pleased to note that the authors have addressed the concerns raised during the initial review process effectively, enhancing the overall quality and depth of the manuscript.

Given the revisions made and the significance of the research findings, I have no further comments or suggestions to add. The manuscript is a valuable addition to the literature on sustainability and low-carbon behaviors, particularly in the context of the millennial generation's role in environmental conservation.

Reviewer 4 Report

Comments and Suggestions for Authors

Bellow are some typs to improve the manuscript.

METHOD

2.2. Sampling

What should be the sample size with 95% of confidence? A sample of 50 participants is quite small to draw any conclusion.

RESULTS

3.1. General Results

Data provided between lines 242 to 250 and 273 to 278 should be organized in a table of statistical description of the participants/repondents.

Reviewer 5 Report

Comments and Suggestions for Authors

I think this article has a good perspective. It may be considered for publication.

Comments on the Quality of English Language

It could have been revised better.